# Evolutionary dynamics of the kinetochore network in eukaryotes as revealed by comparative genomics

Jolien JE van Hooff[1,2,3] (ID), Eelco Tromer[1,2] (ID), Leny M van Wijk[2] (ID), Berend Snel[2,*,†] (ID) & Geert JPL Kops[1,3,4,**,†] (ID)

## Abstract

During eukaryotic cell division, the sister chromatids of duplicated chromosomes are pulled apart by microtubules, which connect via kinetochores. The kinetochore is a multiprotein structure that links centromeres to microtubules, and that emits molecular signals in order to safeguard the equal distribution of duplicated chromosomes over daughter cells. Although microtubule-mediated chromosome segregation is evolutionary conserved, kinetochore compositions seem to have diverged. To systematically inventory kinetochore diversity and to reconstruct its evolution, we determined orthologs of 70 kinetochore proteins in 90 phylogenetically diverse eukaryotes. The resulting ortholog sets imply that the last eukaryotic common ancestor (LECA) possessed a complex kinetochore and highlight that current-day kinetochores differ substantially. These kinetochores diverged through gene loss, duplication, and, less frequently, invention and displacement. Various kinetochore components co-evolved with one another, albeit in different manners. These co-evolutionary patterns improve our understanding of kinetochore function and evolution, which we illustrated with the RZZ complex, TRIP13, the MCC, and some nuclear pore proteins. The extensive diversity of kinetochore compositions in eukaryotes poses numerous questions regarding evolutionary flexibility of essential cellular functions.

**Keywords** co-evolution; eukaryotic diversity; evolutionary cell biology; gene loss; kinetochore

**Subject Categories** Cell Cycle; Evolution; Systems & Computational Biology

## Introduction

During mitotic cell division, eukaryotes physically separate duplicated sister chromatids using microtubules within a bipolar spindle. These microtubules pull the sister chromatids in opposite directions, toward the spindle poles from which they emanate [1]. Current knowledge indicates that all eukaryotes use microtubules for chromosome separation, suggesting that the last eukaryotic common ancestor (LECA) also did. Microtubules and chromatids are connected by the kinetochore, a multiprotein structure that is assembled on the centromeric chromatin [2,3]. Functionally, the kinetochore proteins can be subdivided into three main categories: proteins that connect to the centromeric DNA (inner kinetochore), proteins that connect to the spindle microtubules (outer kinetochore), and proteins that perform signaling functions at the kinetochore in order to regulate chromosome segregation. These signaling functions consist of the spindle assembly checkpoint (SAC), which prevents sister chromatids from separating before all have stably attached to spindle microtubules, and attachment error correction, which ensures that these sister chromatids are attached by microtubules that emanate from opposite poles. Together, the SAC and error correction machineries ensure that both daughter cells acquire a complete set of chromosomes.

Although microtubule-mediated chromosome segregation is conserved across eukaryotes, their mitotic mechanisms differ. For example, some species, such as those in animal lineages, disassemble the nuclear envelope during mitosis ("open mitosis"), while others, such as yeasts, completely or partially maintain it ("(semi-) closed mitosis") [4]. Species differ also in their kinetochore composition, both in the inner and in the outer kinetochore. For example, *Drosophila melanogaster* and *Caenorhabditis elegans* lack most components of the constitutive centromere-associated network (CCAN), a protein network in the inner kinetochore. In the outer kinetochore, diverse species employ either the Dam1 (e.g., various Fungi, Stramenopila, and unicellular relatives of Metazoa) or the Ska complex (most Metazoa and Viridiplantae and some Fungi) for tracking depolymerizing microtubules [5]. The kinetochore of the excavate species *Trypanosoma brucei* mostly consists of proteins that do not seem homologous to the "canonical" kinetochore proteins [6,7]. Studying the evolution of kinetochore proteins revealed how kinetochore diversity was shaped by different modes

1  Hubrecht Institute − KNAW (Royal Netherlands Academy of Arts and Sciences), Utrecht, The Netherlands
2  Theoretical Biology and Bioinformatics, Department of Biology, Science Faculty, Utrecht University, Utrecht, The Netherlands
3  Molecular Cancer Research, University Medical Center Utrecht, Utrecht, The Netherlands
4  Cancer Genomics Netherlands, University Medical Center Utrecht, Utrecht, The Netherlands
  *Corresponding author. Tel: +31 302538102; E-mail: b.snel@uu.nl
  **Corresponding author. Tel: +31 302121907; E-mail: g.kops@hubrecht.eu
  †These authors contributed equally to this work as senior authors

of genome evolution: The inner kinetochore CenpB-like proteins were recurrently domesticated from transposable elements [8], the outer kinetochore protein Knl1 displays recurrent repeat evolution [9], the SAC proteins Bub1/BubR1/Mad3 (MadBub) duplicated and subfunctionalized multiple times in eukaryotic evolution [10,11], and the SAC protein p31[comet] was recurrently lost [12].

Prior comparative genomics studies reported on kinetochore compositions in eukaryotes [12,13]. These studies raised various questions: Are kinetochores in general indeed highly diverse? How often do kinetochore proteins evolve in a recurrent manner in different lineages? How frequent is loss of kinetochore proteins? Does the kinetochore consist of different evolutionary modules? To address these and other questions, we studied the eukaryotic diversity of the kinetochore by scanning a large and diverse set (90) of eukaryotic genomes for the presence of 70 kinetochore proteins. We deduced the kinetochore composition of LECA and shed light on how, after LECA, eukaryotic kinetochores diversified. To understand this evolution functionally, we detected co-evolution among kinetochore complexes, proteins and sequence motifs: Co-evolving kinetochore components are likely functionally interdependent. Furthermore, we found that certain species contain yet inexplicable kinetochore compositions, such as absences of proteins that are crucial in model organisms. We nominate such species for further investigation into their mitotic machineries.

## Results

### Eukaryotic diversity in the kinetochore network

We selected 70 proteins that compose the kinetochore (see Materials and Methods). For comparison, we also included proteins that constitute the anaphase-promoting complex/cyclosome (APC/C), which is targeted by kinetochore signaling. We identified orthologous sequences of these kinetochore and APC/C proteins in 90 diverse eukaryotic lineages by performing in-depth homology searches. Our methods were aimed at maximizing detection of a protein's orthologs even if it evolves rapidly, which is the case for many kinetochore proteins (as we discuss below). The resulting sets of orthologous sequences are available (Dataset EV1). We projected the presences and absences of proteins ("phylogenetic profiles") across eukaryotes (Fig 1, Materials and Methods). In spite of our thorough homology searches, for some proteins the ortholog in a given species might have diverged too extensively to recognize it, resulting in a "false" absence. We however think that, globally, our analysis gives an accurate representation of kinetochore proteins in eukaryotes (Discussion).

We inferred the evolutionary histories of the proteins by applying Dollo parsimony, which allows only for a single invention and infers subsequent losses based on maximum parsimony. Of the 70 kinetochore proteins, 49 (70%) were inferred to have been present in LECA (Figs 1 and 2A and C). CenpF, Spindly and three subunits of the CenpO/P/Q/R/U complex probably originated more recently. The Dam1 complex likely originated in early fungal evolution and may have propagated to non-fungal lineages via horizontal gene transfer [5].

Kinetochore proteins are less conserved than APC/C subunits (Fig EV1, Appendix Table S1, [14]). Species on average possess 48% of the kinetochore proteins, compared to 70% of the APC/C subunits. Species that we predict to contain relatively few kinetochore proteins include *Tetrahymena thermophila* (Fig 2B) and *Cryptococcus neoformans* (Fig 2D). Some kinetochore proteins are absent from many different lineages, likely resulting from multiple independent gene loss events. We counted losses of kinetochore and APC/C proteins during post-LECA evolution using Dollo parsimony. On average, kinetochore proteins were lost 16.5 times since LECA, while APC/C proteins were lost 13.1 times (not significantly different for kinetochore versus APC/C). Our homology searches hinted at some kinetochore proteins evolving also rapidly on the sequence level. The kinetochore proteins indeed have relatively high dN/dS values, a common measure for sequence evolution: When comparing mouse and human gene sequences, kinetochore proteins scored an average dN/dS of 0.24, compared to 0.06 for the APC/C proteins ($P = 0.0016$) and 0.15 for all human proteins ($P = 4.8\text{e-}5$). The loss frequency and sequence evolution seem to be correlated, suggesting a common underlying cause for poor conservation (Fig EV2, Discussion). Overall, the kinetochore seems to evolve more flexibly than the APC/C.

We not only mapped the presences and absences of kinetochore proteins, but also counted their copy number in each genome (Fig EV3). As observed before, MadBub and Cdc20 are often present in multiple copies. These proteins likely duplicated in different lineages and subsequently the resulting paralogs subfunctionalized [10–12]. CenpE, Rod, Survivin, Sgo and the mitotic kinases Aurora and Plk also have elevated copy numbers. Possibly, these proteins also underwent (recurrent) duplication and subfunctionalization, as, for example, suggested for Sgo: In the lineages of *Schizosaccharomyces pombe*, *Arabidopsis thaliana* and mammals, Sgo duplicated and likely subsequently subfunctionalized in a recurrent manner [15–17].

### Co-evolution within protein complexes of the kinetochore

Subunits of a single kinetochore complex tend to co-occur across genomes: They have similar patterns of presences and absences ("phylogenetic profiles", Fig 1A). Such co-occurring subunits likely co-evolved as a functional unit [18]. To quantify how similar phylogenetic profiles are, we calculated the Pearson correlation coefficient ($r$) for each kinetochore protein pair. We defined a threshold of $r = 0.477$ for protein pairs likely to be interacting, based on the scores among established interacting kinetochore pairs (Appendix Fig S1). All pairwise scores were used to cluster the proteins (Fig 1 including Datasets EV1 and EV2) and to visualize the proteins using t-Distributed Stochastic Neighbor Embedding (t-SNE, Appendix Fig S2) [19]. Many established interacting proteins correlate well and, as a result, cluster together and are in close proximity in our t-SNE map. Examples include the SAC proteins Mad2 and MadBub, centromere proteins (CENPs) located in the inner kinetochore (discussed below), the Ska complex and the Dam1 complex. Such complexes, with subunits having highly similar phylogenetic profiles, evolved as a functional unit.

While some kinetochore proteins have highly similar phylogenetic profiles, others lack similarity, pointing to a more complex interplay between evolution and function. First, two proteins might have strongly dissimilar, or inverse, phylogenetic profiles, potentially because they are functional analogs [20]. In the kinetochore

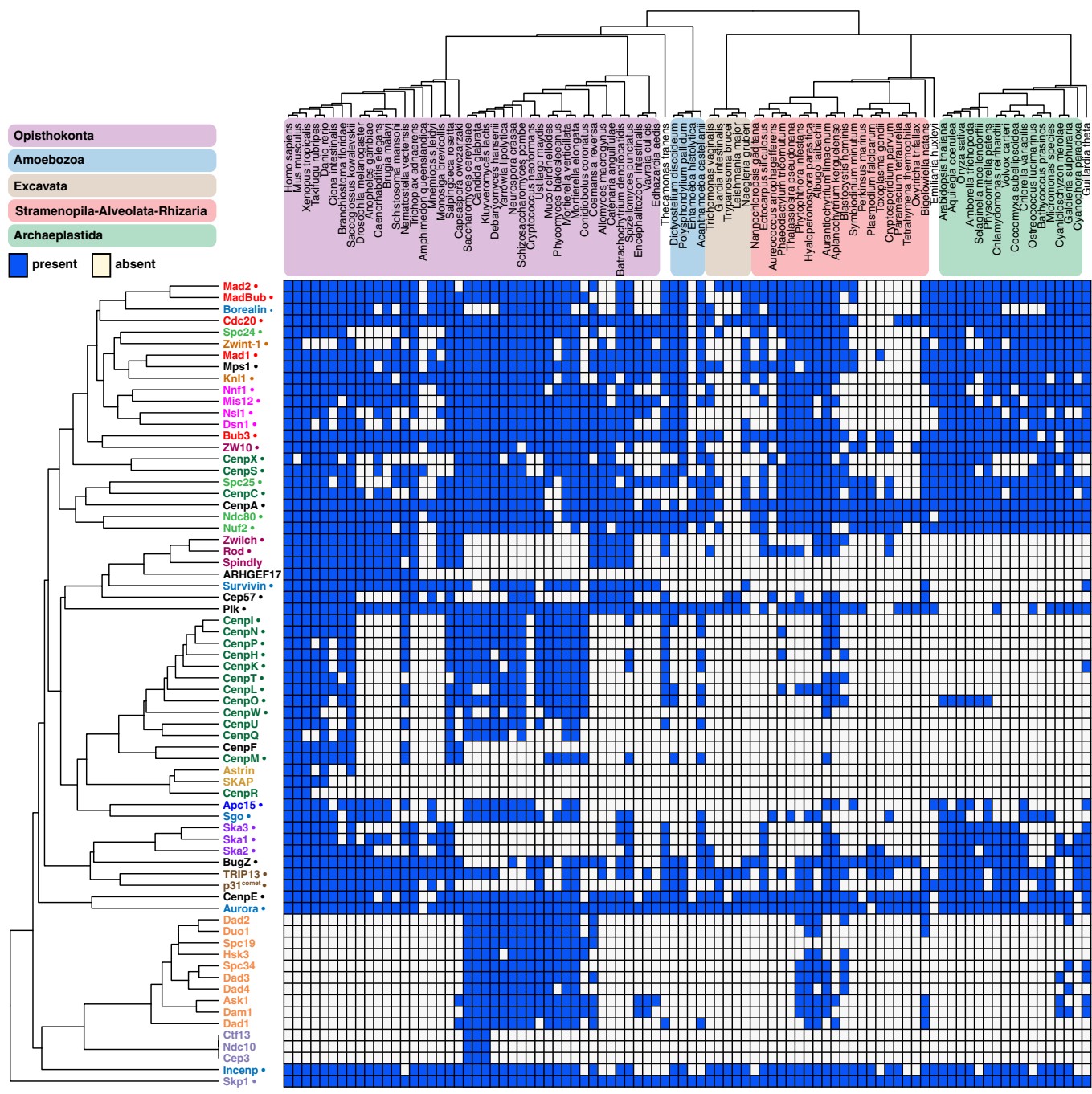

**Figure 1. The kinetochore network across 90 eukaryotic lineages.**

Presences and absences ("phylogenetic profiles") of 70 kinetochore proteins in 90 eukaryotic species. Top: Phylogenetic tree of the species in the proteome set, with colored areas for the eukaryotic supergroups. Left side: Kinetochore proteins clustered by average linkage based on the pairwise Pearson correlation coefficients of their phylogenetic profiles. Protein names have the same colors if they are members of the same complex. Proteins inferred to have been present in LECA are indicated (●). The orthologous sequences (including sets of APC/C subunits, NAG, RINT1, HORMAD, Nup106, Nup133, Nup160) are available as fasta files in Dataset EV1, allowing full usage of our data for further evolutionary cell biology investigations.

network, phylogenetic dissimilarity is observed for proteins of the Dam1 complex and of the Ska complex, which are indeed analogous complexes [5,21,22]. Second, proteins that do interact in a complex might nevertheless have little similarity in their phylogenetic profiles. Either such a complex did not evolve as a functional unit because its subunits started to interact only recently [23], or because

one of its subunits serves a non-kinetochore function and thus also co-evolves with non-kinetochore proteins [24]. An example of a potentially recently emerged interaction is BugZ-Bub3, that form a kinetochore complex in human [25,26], but have little similarity in their phylogenetic profiles, measured by their low correlation ($r = 0.187$). In general, BugZ's phylogenetic profile is different from

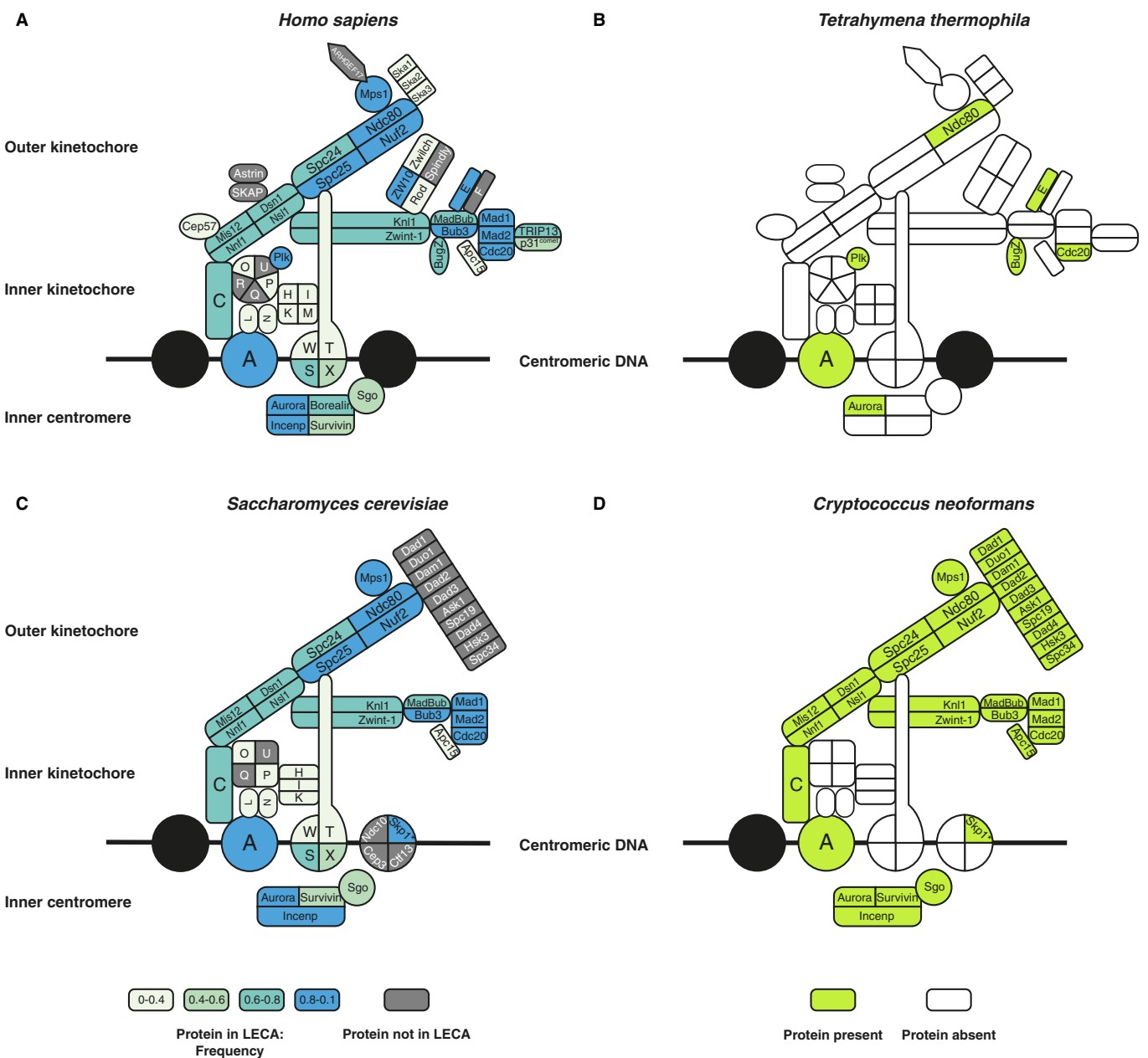

**Figure 2.  Kinetochores of model and non-model species.**

A  The human kinetochore. The colors of the proteins indicate if they were inferred to be present in LECA and their occurrence frequency across eukaryotes (see Materials and Methods).

B  The predicted kinetochore of *Tetrahymena thermopila* projected onto the human kinetochore.

C  The budding yeast kinetochore. Similar to panel (B).

D  The predicted kinetochore of *Cryptococcus neoformans* projected onto the budding yeast kinetochore.

other kinetochore proteins; hence, this protein might be recently added to the kinetochore [27,28]. An example of a kinetochore protein that co-evolves with non-kinetochore proteins is ZW10, which joins Rod and Zwilch in the RZZ complex. The phylogenetic profile of ZW10 is dissimilar from those of Rod and Zwilch ($r = 0.218$ for Rod, $r = 0.236$ for Zwilch), while those are very similar to each other ($r = 0.859$, Fig 3), due to ZW10 being present in various species that lack Rod and Zwilch. In those species, ZW10

might not localize to the kinetochore but perform only in vesicular trafficking, in a complex with NAG and RINT1 (NRZ complex [29]). Indeed, the ZW10 phylogenetic profile is much more similar to that of NAG ($r = 0.644$) and RINT1 ($r = 0.512$) compared to Rod and Zwilch. Hence, ZW10 more strongly co-evolves with NAG and RINT1. The Rod and Zwilch phylogenetic profiles are similar to that of Spindly ($r = 0.730$ for Rod, $r = 0.804$ for Zwilch), a confirmed RZZ-interacting partner [30–32]. These similarities argue for an

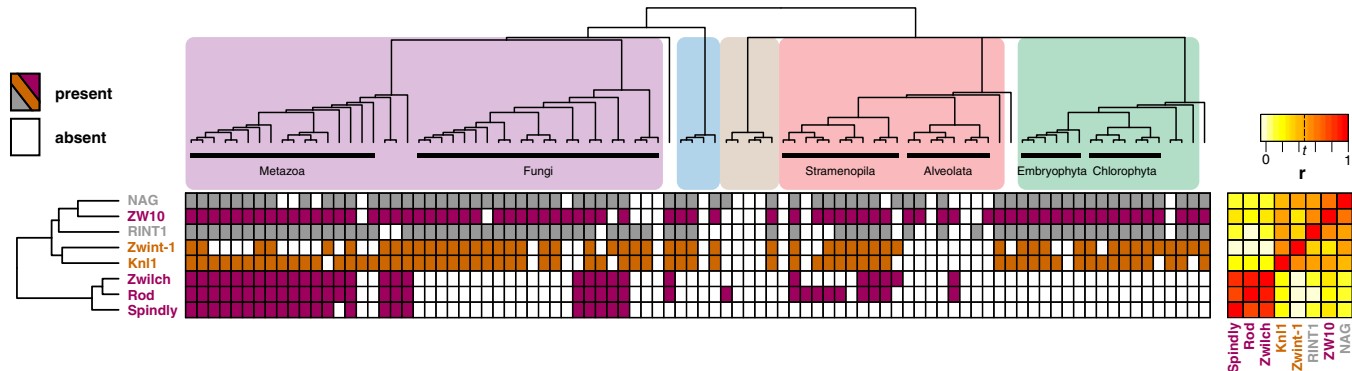

**Figure 3.  Phylogenetic profiles of the Rod–Zwilch–ZW10 (RZZ) complex, its mitotic interaction partners (Knl1, Zwint-1, and Spindly), and ZW10's interphase interaction partners in the NRZ (NAG and RINT1) complex.**

Presences and absences across eukaryotes of the RZZ subunits, Spindly, Zwint-1, and Knl1, and of the NRZ subunits, NAG and RINT1. Colored areas indicate eukaryotic supergroups as in Fig 1. Right side: Pairwise Pearson correlation coefficients (*r*) between the phylogenetic profiles including a heatmap. The indicated threshold *t* represents the value of *r* for which we found a sixfold enrichment of interacting protein pairs (see Appendix Fig S1). See also Appendix Fig S3 for the procedure by which homology between Zwint-1, Sos7, and Kre28 was detected.

evolutionary "Rod–Zwilch–Spindly" (RZS) module, rather than an RZZ module.

The phylogenetic profiles of kinetochore proteins shed new light on these proteins' (co-)evolution and on their function, examples of which are discussed in detail below.

### The CCAN evolved as an evolutionary unit that is absent from many lineages

The kinetochore connects the centromeric DNA, mainly via CenpA, to the spindle microtubules, mainly via Ndc80. In human and yeast, CenpA and Ndc80 are physically linked via the constitutive centromere-associated network (CCAN, reviewed in [33]). Physically, the CCAN comprises multiple protein complexes (Fig 2). Evolutionarily, however, it comprises a single unit, as the majority of CCAN proteins have highly similar phylogenetic profiles (Fig 1, average $r = 0.513$). Four CCAN proteins are very different from the others: CenpC, CenpR, CenpX, and CenpS. CenpC is widely present and is sufficient to assemble at least part of the outer kinetochore in *D. melanogaster* and humans [34,35]. CenpR seems a recent gene invention in animals. CenpX and CenpS have a more ubiquitous distribution compared to other CCAN proteins, possibly due to their non-kinetochore role in DNA damage repair [36,37].

Our study confirmed that most CCAN proteins have no (detectable) homologs in *C. elegans* and *D. melanogaster*. The CCAN is not only absent from these model species, but also from many other lineages, such as various animals and fungi, and all Archaeplastida. Because the CCAN is found in three out of five eukaryotic supergroups, it likely was present in LECA, and subsequently lost multiple times in diverse eukaryotic lineages. Alternatively, the CCAN was invented more recently and horizontally transferred among eukaryotic supergroups. However, under both scenarios the CCAN was recently lost in various lineages, for example in the basidiomycete fungi: While *Ustilago maydis* has retained the CCAN, its sister clade *C. neoformans* eliminated it (Fig 2D). The finding that most of the CCAN (with the exception of CenpC) is absent in many

eukaryotic lineages poses questions about kinetochore architectures in these species. Since they generally possess a protein binding to the centromeric DNA (CenpA, see Fig EV4 for details on identifying the orthologs of CenpA) and a protein binding to the spindle microtubules (Ndc80), their kinetochore is not wholly unconventional. Is the bridging function of the CCAN simply dispensable, as proposed for *D. melanogaster* [38], or is it carried out by other, non-homologous protein complexes? In order to answer these questions, the kinetochores of diverse species that lack the CCAN should be experimentally examined in more detail.

### Absence of co-evolution between RZS and its putative kinetochore receptor Zwint-1

Various studies suggested that the RZZ/RZS complex is recruited to the kinetochore primarily by Zwint-1. Zwint-1 itself localizes to the kinetochore by binding to Knl1 [39,40]. We compared the phylogenetic profile of Zwint-1 to the profiles of these interaction partners: RZZ/RZS and Knl1 (Fig 3). While we searched for orthologs of Zwint-1, we concluded that Zwint-1, Kre28 (*S. cerevisiae*), and Sos7 (*S. pombe*) belong to the same orthologous group [41,42], collectively referred to as "Zwint-1". Although these sequences are only weakly similar, they can be linked by multidirectional homology searches (Appendix Fig S3).

Our set of 90 species contains many species that possess a Zwint-1 ortholog (36 species) but lack RZS, and vice versa (11 species, $-0.065 < r < 0$). This lack of correlation strongly suggests that, at least in a substantial amount of lineages, RZZ/RZS is not recruited to kinetochores by Zwint-1, but by another, yet unidentified factor. Support for this inference was recently presented in studies using human HeLa cells [43,44]. Compared to RZS, the phylogenetic profile of Zwint-1 is more similar to that of Knl1 (Fig 3, $r = 0.506$), and of Spc24 and Spc25 (Fig 1, $r = 0.529$ for Spc24, $r = 0.499$ for Spc25), two subunits of the Ndc80 complex that are located in close proximity to Knl1-Zwint-1 [45]. Perhaps Zwint-1 stabilizes the largely unstructured protein Knl1 [44], thereby indirectly affecting the recruitment of RZZ/RZS.

## Higher-order co-evolution between the AAA+ ATPase TRIP13 and HORMA domain proteins

SAC activation and SAC silencing are both promoted by the AAA+ ATPase TRIP13. TRIP13 operates by using the HORMA domain protein p31$^{comet}$ to structurally inactivate the SAC protein Mad2, also a HORMA domain protein (Fig 4A). Since the SAC requires Mad2 to continuously cycle between inactive and active conformations, TRIP13 enables SAC signaling in prometaphase. In metaphase, however, when no new active Mad2 is generated, TRIP13 stimulates SAC silencing [46–48]. The TRIP13 ortholog of budding yeast, Pch2, probably has a molecularly similar function in meiosis: Pch2 is proposed to bind oligomers of the HORMA domain protein Hop1 (HORMAD1 and HORMAD2 in mammals, hereafter referred to as "HORMAD") and to structurally rearrange one copy within the oligomer, resulting in its redistribution along the chromosome axis. HORMAD, p31$^{comet}$ and Mad2 are homologous as they belong to the family of HORMA domain proteins that also includes Rev7 [49] and autophagy-related proteins Atg13 and Atg101 [50,51]. All of these proteins likely descend from an ancient HORMA domain protein that duplicated before LECA.

Although the TRIP13 phylogenetic profile is relatively similar to both the profiles of p31$^{comet}$ ($r = 0.526$) and HORMAD ($r = 0.517$), TRIP13 does not co-occur with these proteins in multiple species (Fig 4B). These exceptions to the co-occurrences of TRIP13/p31 and TRIP13/HORMAD can be explained by the dual role of TRIP13, which is to interact with both p31$^{comet}$ and with HORMAD. If we combine profiles of p31$^{comet}$ and HORMAD, the similarity with TRIP13 increases: the joint p31$^{comet}$ and HORMAD profile strongly correlates with the TRIP13 profile ($r = 0.766$, Fig 4C). TRIP13 was indeed expected to co-evolve with both of its interaction partners, as has been demonstrated for other multifunctional proteins [24]. Based on the phylogenetic profiles, we conclude that TRIP13 is only retained if at least p31$^{comet}$ or HORMAD is present (with the exception of the diatom *Phaeodactylum tricornutum*). We predict that TRIP13-containing species that lost p31$^{comet}$ but retained HORMAD, such as *S. cerevisiae* and *Acanthamoeba castellanii*, only use TRIP13 during meiosis and not in mitosis.

## The phylogenetic profiles of SAC proteins predict a role for nuclear pore proteins in the SAC

Because similar phylogenetic profiles reflect the functional interaction of proteins, similar phylogenetic profiles also predict such interactions. We applied this rationale by comparing the phylogenetic profiles of the kinetochore proteins (Fig 1) to those of proteins of the genome-wide PANTHER database in search of unidentified connections. PANTHER is a database of families of homologous proteins from complete genomes across the tree of life. We assigned all proteins present in our eukaryotic proteome database to these homologous families (see Materials and Methods). For each kinetochore protein in Fig 1, we listed the 30 best matching (with the highest Pearson correlation coefficient) families in PANTHER, and screened for PANTHER families that occur often in these lists (Appendix Table S3). Of these families, we considered the nuclear pore protein Nup160 an interesting hit, because it is part of the Nup107-Nup160 nuclear pore complex that localizes to the kinetochore [52,53]. The phylogenetic profile of Nup160 (as defined by PANTHER) was particularly similar to that of the SAC protein MadBub ($r = 0.718$). In order to improve the phylogenetic profile of Nup160, we manually determined the orthologous group of Nup160 in our own proteome dataset. We also determined those of Nup107 and Nup133, two other proteins of the Nup107-Nup160 complex. The Nup160, Nup133 and Nup107 phylogenetic profiles strongly correlated with those of SAC proteins MadBub ($0.541 < r < 0.738$) and Mad2 ($0.528 < r < 0.715$, Fig 5)—even stronger than these three nuclear pore proteins correlated with one another ($0.475 < r < 0.601$). Furthermore, Nup160, Nup133 and Nup107 correlate better with MadBub and Mad2 than these SAC proteins do with the other SAC proteins (MadBub: average $r = 0.563$, Mad2: average $r = 0.511$) and far better than these SAC proteins do with all kinetochore proteins (MadBub: average $r = 0.290$, Mad2: average $r = 0.239$). While previous studies have shown that the Nup107-Nup160 complex localizes to the kinetochore in mitosis, our analysis in addition suggests that these proteins may function in the SAC and that they potentially interact with Mad2 and MadBub.

## The Mad2-interacting motif (MIM) in Mad1 and Cdc20 is coupled with Mad2 presence

While interacting proteins are expected to co-evolve at the protein–protein level, as exemplified by many complexes within the kinetochore, interacting proteins might also co-evolve at different levels, such as protein-motif. Co-evolution between a protein and a sequence motif has been incidentally detected before, for example in case of CenpA and its interacting motif in CenpC [54] and in case of MOT1 and four critical phenylalanines in TBP [55]. We here explore potential co-evolution of Mad2 with the sequence motif it interacts with in Cdc20 and Mad1: the Mad2-interacting motif (MIM). Both the Mad2–Mad1 and the Mad2–Cdc20 interactions operate in the SAC [56,57]. We defined the phylogenetic profiles of the MIM in Mad1 and Cdc20 [58,59] (Fig 6A) by inspecting the multiple sequence alignments of Mad1 and Cdc20. These alignments revealed that the MIM is found at a similar position across the Mad1 and Cdc20 orthologs; hence, the motif likely predates LECA in both these proteins. Notable differences exist between the MIMs of

**Figure 4.  The co-evolutionary patterns of the multifunctional protein TRIP13.**

A   Model for the mode of action of TRIP13 as recently suggested [79]. By hydrolyzing ATP, TRIP13 changes the conformation of HORMAD and Mad2 from closed to open, the latter via binding to co-factor p31$^{comet}$, which forms a heterodimer with Mad2. TRIP13 has a C-terminal AAA$^+$ ATPase domain (AAA$^+$) and an N-terminal domain (NTD) and forms a hexamer [80].

B   Presences and absences of TRIP13 and of its interaction partners p31$^{comet}$ and HORMAD. Colored areas indicate eukaryotic supergroups as in Fig 1.

C   Numbers of lineages in which TRIP13 is present or absent, compared to the presences of p31$^{comet}$, HORMAD or their joint presences. Also the Pearson correlation coefficients of the phylogenetic profiles as in (B) are given.

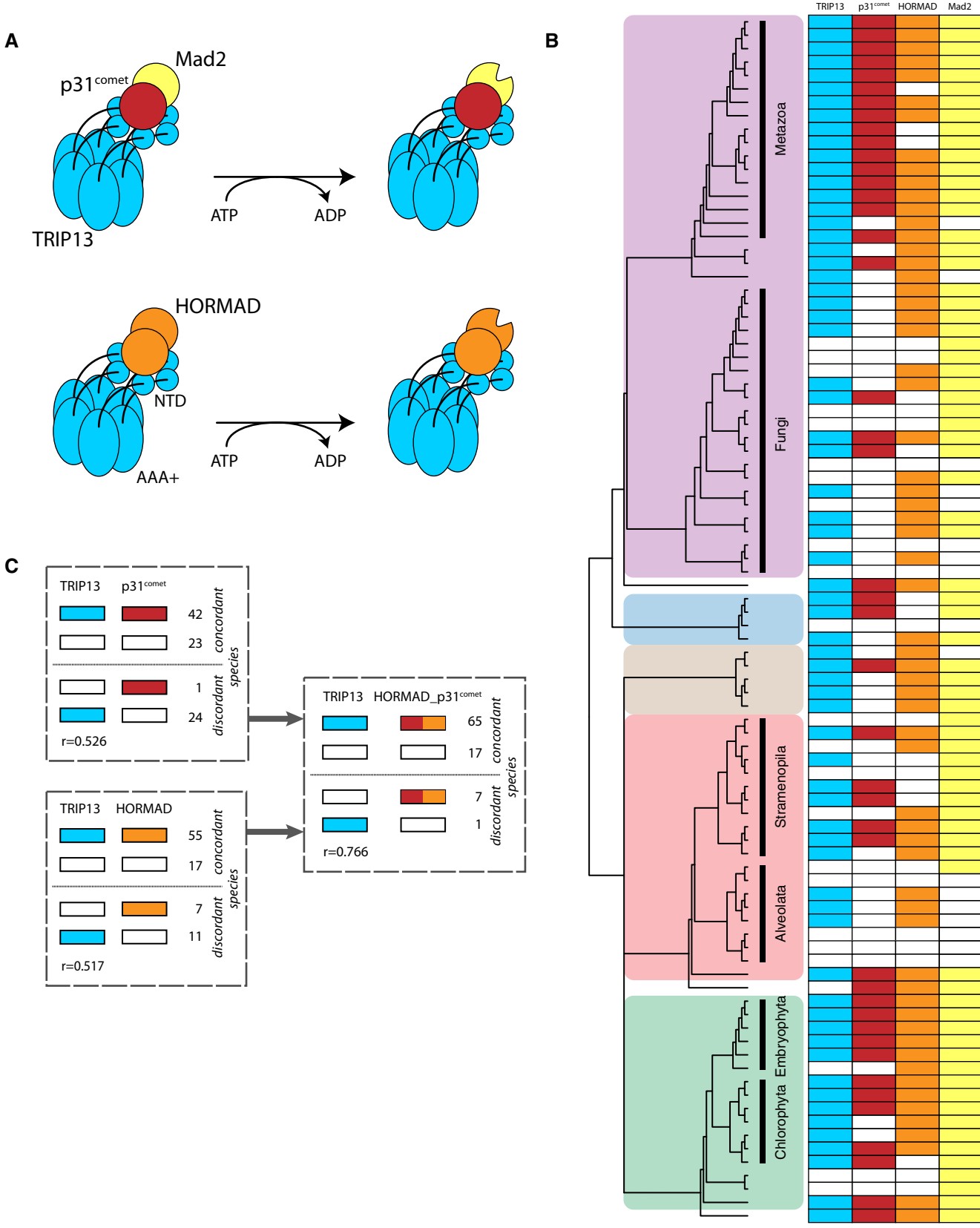

**Figure 4.**

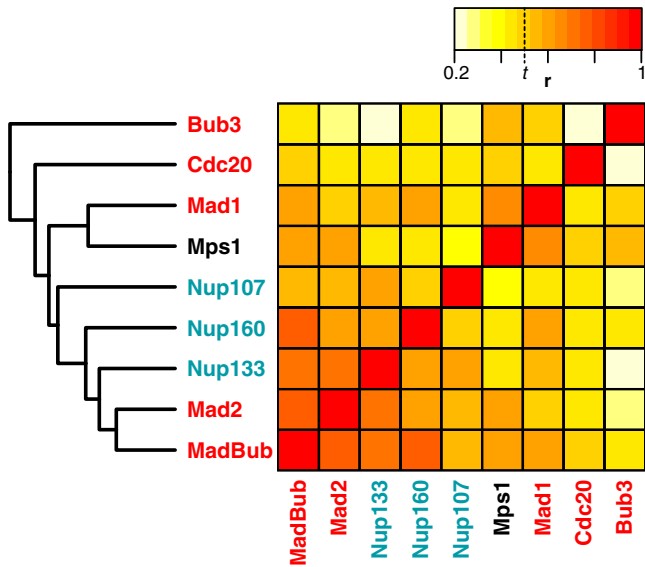

**Figure 5. Correlations between proteins of the Nup107-160 complex and proteins of the SAC.**

Heatmap indicating the pairwise Pearson correlation coefficients (*r*) of the phylogenetic profiles of proteins of the Nup107-160 complex and of the SAC. The clustering (average linkage) on the left side of this heatmap was also based on these correlations. The indicated threshold *t* represents the Pearson correlation coefficient for which we found a sixfold enrichment of interacting protein pairs (see Appendix Fig S1).

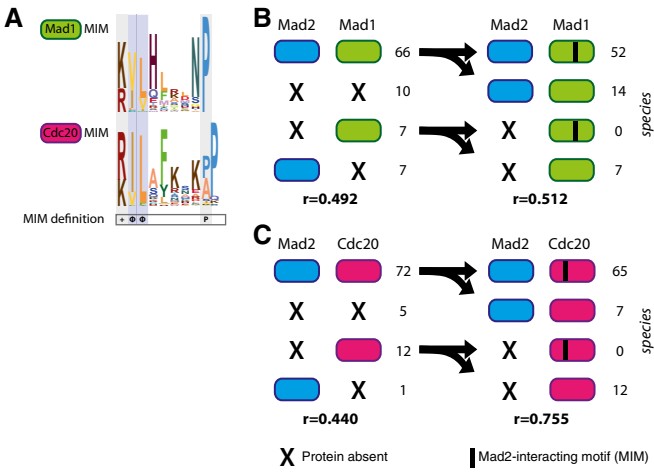

**Figure 6. Phylogenetic co-occurrence of Mad2 with its interaction partners Mad1 and Cdc20 and their Mad2-interacting motifs (MIMs).**

A    The sequence logos of the MIMs of Mad1 (upper panel) and Cdc20 (lower panel) based on the multiple sequence alignments of the motifs. Below is indicated the required amino acid sequence of the MIM (+: positive residue, Φ: hydrophobic residue, P: proline) which is restricted by the pattern [RK] [ILV](2)X(3,7)P.

B, C    Left side: Numbers of presences and absences of Mad2 in 90 eukaryotic species and its interaction partners Mad1 (B) and Cdc20 (C). Right side: Frequencies of Mad2 and canonical MIM occurrences in species having Mad1 (B) or Cdc20 (C), respectively. Also the Pearson correlation coefficients (*r*) for the corresponding phylogenetic profiles are shown.

Cdc20 and Mad1, which could reflect differences in binding strength to Mad2.

The phylogenetic profiles of Mad2 and of the MIM in Cdc20 or Mad1 orthologs correlated stronger than the full-length proteins (Fig 6B and C). In particular, species lacking Mad2, but having Mad1 and/or Cdc20, never contained the canonical MIM in either their Cdc20 or their Mad1 sequences (hypergeometric test: $P < 10^{-4}$, $P < 10^{-9}$ for Mad1 and Cdc20, respectively). Such species hence likely lost Mad2 and subsequently lost the MIM in Mad1 and Cdc20, because it was no longer functional. Moreover, absence of the MIM in Mad1/Cdc20 supports that in these species Mad2 is indeed absent. While we expected to only find a MIM in species that actually have Mad2, we also expected the reverse: that species that have Mad2 also have a MIM in their Mad1/Cdc20. This is however not the case, most notably for Mad1: Many lineages (14) have both Mad1 and Mad2 but lack the MIM in Mad1. A substantial fraction (six) of this group belongs to the land plant species that have a somewhat different motif in Mad1 that is conserved within this lineage (Fig EV5A). This altered land plant motif might mediate the Mad1–Mad2 interaction, which has been reported in *A. thaliana* [60]. If we consider this plant motif to be a "valid" MIM, the Mad1-MIM and Mad2 correlate substantially better (Fig EV5B–D). Overall, under both motif definitions the protein-motif correlations are higher than the protein–protein correlations. Hence, including sequence motifs can expose that interaction partners co-evolve, albeit at a different level, and may aid to predict functional interactions between proteins *de novo*.

## Discussion

Our evolutionary analyses revealed that since LECA, the kinetochores of different lineages strongly diverged by different modes of genome evolution: kinetochore proteins were lost, duplicated and/or invented, or diversified on the sequence level. In addition to straightforward protein–protein co-evolution, we found alternative evolutionary relationships between proteins that hint at a more complex interplay between evolution and function. Some established interacting proteins have not co-evolved (Zwint-1 and RZS, Bub3 and BugZ), which has been previously shown for other interaction partners to reflect evolutionary flexibility [23]. Lack of co-evolution may also reflect that a protein has multiple different functions, for which it interacts with different partners. The phylogenetic profile of such a multifunctional protein differs from either of its interaction partners, and instead is similar to the combined profiles of its interaction partners [24], as we showed for TRIP13 with HORMAD and p31$^{comet}$. Some co-evolutionary relationships predicted novel protein functions, such as nuclear pore proteins operating in the SAC, which should be confirmed with experiments. Finally, not only proteins, but also functional sequence motifs co-evolved with their interaction partner, as we found for the MIMs in Cdc20/Mad1 with Mad2. Probably, including more proteins and (known and *de novo* predicted) motifs/domains will not only improve the correlation between known interaction partners, but will also enhance predicting yet unknown interactions and functions.

While we carefully curated the orthologous groups of each of the kinetochore proteins, their phylogenetic profiles might

contain some false positives and/or false negatives: incorrectly assigned presences (because a protein sequence in fact is not a real ortholog) and incorrectly assigned absences (because a species does contain an ortholog, but we did not detect it). For most kinetochore proteins, we estimate the chance of false negatives larger than of false positives, mainly because they likely are vulnerable to homology detection failure, given that their sequences evolve so rapidly (Appendix Table S1, Results). Such false negatives of a particular protein will result in falsely inferred gene loss events. A failure to detect homology might therefore also cause sequence divergence to correlate with loss frequency (Fig EV2). Specific examples of suspicious absences (potential false negatives) include the inner centromere protein Borealin in *S. cerevisiae* and the KMN network proteins Spc24, Spc25, Nsl1/Dsn1 in *D. melanogaster* and *C. elegans*, and possibly Ndc80 in *T. brucei*, since functional counterparts of these proteins have been characterized in these species [7,61–66]. Moreover, species that we predicted to have very limited kinetochore compositions, such as *T. thermophila* (Fig 2B), might actually contain highly divergent orthologs that we could not detect. If such a species' kinetochore would be examined biochemically, its undetected orthologs might be uncovered. Although the phylogenetic profiles of the kinetochore proteins presented here might contain some of such errors, we think that our manual curation of the orthologs groups (see Materials and Methods) yields an accurate global representation of the presences and absences of these proteins among eukaryotes. We think this accuracy is supported by the high similarity of phylogenetic profiles of interacting proteins.

The set of kinetochore proteins we studied here is strongly biased toward yeast and animal lineages, lineages that are relatively closely related on the eukaryotic tree of life. This bias is due to the extensive experimental data available for these lineages. Highly different kinetochores might exist, such as the kinetochore of *T. brucei* [6,7]. If in the future we know the experimentally validated kinetochore compositions of a wider range of eukaryotic species, we could sketch a more complete picture of kinetochore evolution and could potentially expand and improve our functional predictions.

Since the kinetochore seems highly diverse across species, several questions arise. Is the kinetochore less conserved than other core eukaryotic cellular systems/pathways, as comparing it to the APC/C suggested? And if so, why is it allowed to be less conserved, or are many of the alterations adaptive to the species? Why do certain lineages (such as multicellular animals and plants) contain a particular kinetochore submodule (such as the Ska complex) while others lack it, or have an alternative system (such as Dam1)? Do these genetic variations among species have functional consequences for kinetochore-related processes in their cells? To answer such questions, our dataset should be expanded with specific (cellular) features and lifestyles, when this information becomes available for the species in our genome dataset. Together with biological and biochemical analyses of processes in unexplored species, an expanded dataset may reveal the true flexibility of the kinetochore in eukaryotes and show how chromosome segregation is executed in diverse species. The comparative genomics analysis that we presented here provides a starting point for such an integrated approach into studying kinetochore diversity and evolution, since it allows for informed decisions about which species to study.

# Materials and Methods

### Constructing the proteome database

To study the occurrences of kinetochore genes across the eukaryotic tree of life, we constructed a database containing the protein sequences of 90 eukaryotic species. This size was chosen because we consider it to be sufficiently large to represent eukaryotic diversity, but also sufficiently small to allow for manual detection of orthologous genes. We selected the species for this database based on four criteria. First, the species should have a unique position in the eukaryotic tree of life, in order to obtain a diverse set of species. Second, if available we selected two species per clade, which facilitates the detection of homologous sequences and the construction of gene phylogenies. Third, widely used model species were preferred over other species. Fourth, if multiple proteomes and/or proteomes of different strains of a species were available, the most complete one was selected. Completeness was measured as the percentage of core KOGs (248 core eukaryotic orthologous groups [67]) found in that proteome. If multiple splice variants of a gene were annotated, the longest protein was chosen. A unique protein identifier was assigned to each protein, consisting of four letters and six numbers. The letters combine the first letter of the genus name with the first three letters of the species name. The versions and sources of the selected proteomes can be found in Appendix Table S2.

### Ortholog detection

The set of kinetochore proteins we studied were selected based on three criteria: (i) localizing to the kinetochore, (ii) being present in at least three lineages, and (iii) having an established role, supported by multiple studies, in the kinetochores and/or kinetochore signaling in human or in budding yeast. We applied a procedure comprising two different methods to find orthologs for the kinetochore proteins in our set within our database of 90 eukaryotic proteomes, and the same procedure was followed for determining orthologs of the APC/C proteins, NAG, RINT1, Nup107, Nup133, Nup160, and HORMAD. The method of choice depended on whether or not it was straightforward to find homologs across different lineages for a specific protein. In both methods, initial searches started with the human sequence, or, if the protein is not present in humans, with the budding yeast sequence.

Method 1. If many homologs were easily found, the challenge was to distinguish orthologs from outparalogs. Here, we defined an orthologous group as comprised of proteins that result from speciation events and that can be traced back to a single gene in LECA, whereas outparalogs are related proteins that resulted from a pre-LECA duplication. For example, Cdc20 and Cdh1 are homologous proteins, both having their own orthologous groups among the eukaryotes. They resulted from a duplication before LECA; therefore, members of the Cdc20 and Cdh1 group are outparalogs to each other. To find homologs, we used blastp online to search through the non-redundant protein sequences (nr) as a database [68]. We aligned the sequences found with MAFFT [69] (version v7.149b,

option einsi, or linsi in case of expected different architectures) to make a profile HMM (www.hmmer.org, version HMMER 3.1b1). If the homologs are known to share only a certain domain, that domain was used for the HMM; otherwise, we used the full-length alignment. This HMM was used as input for hmmsearch to detect homologs across our own database of 90 eukaryotic proteomes. From the hits in this database, we took a substantial number of the highest scoring hit sequences, up to several hundreds. We aligned the hit sequences using MAFFT and trimmed the alignment with trimAl [70] (version 1.2, option automated1). Subsequently, RAxML version 8.0.20 [71] was used to build a gene tree (settings: varying substitution matrices, GAMMA model of rate heterogeneity, rapid bootstrap analysis of 100 replicates). We interpreted the resulting gene tree by comparing it to the species tree and thereby determined which clusters form orthologous groups. These orthologous groups were identified by finding the cluster that contained sequences from a broad range of eukaryotic species and had a sister cluster that also has sequences from this broad range of species. The cluster that contained the initial human query sequence was the orthologous group of interest, while the sister cluster is the group of outparalogs. In our search of orthologs of CenpA, we applied this first method. CenpA is part of the large family of histone H3 proteins and has long been recognized to diverge rapidly, due to which it is a challenge to reconstruct CenpA's evolution [72]. We determined this orthologous cluster with help of experimentally identified centromeric histone H3 variants in a wide range of species, and we included two *Toxoplasma gondii* sequences that were not part of this cluster. For details, see Fig EV4. The tree in this figure was visualized using iTOL [73].

Method 2. If homologs were not easily found, no outparalogs were obtained by these searches and hence the homologs defined the orthologous group. For these cases, we used a different strategy to find the orthologous group in our database. Iterative searching methods (jackhmmer and/or psi-blast) were applied to find homologs across the nr and UniProt database [74]. In specific cases, we cut the initial query sequence, for example to remove putative coiled-coil regions. If a protein returned very few hits, we tried to expand the set of putative homologous sequences by using some of the initially obtained hits as a query. If candidate orthologous proteins were reported in experimental studies in species other than human or budding yeast, but not found by initial searches, we specifically searched using those as a query. If this search yielded hits overlapping with previous searches, these candidate orthologous sequences were added to the set of hits. The sequences in this set were aligned to obtain a refined profile HMM. In addition, we searched for conserved motifs in the hit sequences using MEME [75] (version 4.9.0), which aided in recognizing conserved positions that could characterize the homologs. The obtained profile HMM was used to search for homologs across in local database. The resulting hits were checked for the motifs identified by MEME and applied to online (iterative) homology searches to check whether we retrieved sequences already identified as orthologous. Based on this evaluation of individual hits, we defined a scoring threshold for the hmmsearch with this profile HMM and searched our database until no new hits were found. The resulting set of sequences was the orthologous group of interest. The sequences of the orthologous groups can be found in the Dataset EV1.

## Calculating correlations between phylogenetic profiles

In order to study the co-evolution of the kinetochore proteins and to infer potential functional relationships of these genes based on co-evolution, we derived the phylogenetic profiles of these genes. The phylogenetic profile of a gene is a list of its presences and absences across our set of 90 eukaryotic genomes based on the composition of the orthologous groups. The phylogenetic profile consists of a string of 90 characters containing a "1" if the gene is present in a particular species (either single- or multicopy), and a "0" if it is absent. To reveal whether two genes often co-occur in species, we measured how similar their phylogenetic profiles were using the Pearson correlation coefficient [76]. All pairwise scores can be found in Dataset EV2. To identify pairs of proteins that potentially have a functional association, we applied a threshold of $r = 0.477$. Appendix Fig S1 clarifies why the Pearson correlation coefficient was opted for and how the threshold was set. The Pearson correlation coefficients of all gene pairs were converted into distances ($d = 1 - r$), and the genes were clustered based on their phylogenetic profiles using average linkage. The Pearson correlation coefficients were also used to map the kinetochore proteins in 2D by Barnes-Hut t-SNE (Appendix Fig S2) [19].

## Detecting the MIM in Mad1 and Cdc20 orthologs

We made multiple sequence alignments of the Cdc20 and Mad1 orthologous groups using MAFFT (option einsi). We used these alignments to search for the Mad2-interacting motif (MIM). The typical MIM is defined by [KR] [IVL](2)X(3,7)P for both Mad2 and Cdc20 [58,59], but we also used an alternative definition: [ILV](2)X (3,7)P or [RK][ILV](2). We inferred that the location of the motif in the protein is conserved in Mad2 as well as in Cdc20, because the position of the MIM in the multiple sequence alignments was the same in highly divergent species (e.g., plants and animals). For all orthologous sequences, we checked whether the motif, either the typical MIM (Fig 6) or the alternative MIM (Fig EV5) was present on these conserved positions.

## Finding novel proteins functioning in the kinetochore

To find new proteins performing essential roles at the kinetochore by phylogenetic profiling, a reference protein set was needed. This reference set was based on the protein families present in PANTHER. More specifically, we assigned the proteins within our proteome database of 90 eukaryotic species to PANTHER (sub)families [77] (version 10). This assignment was done by applying hmmscan to the protein sequences of our database, using the complete set of PANTHER family and subfamily HMMs as a search database. Each protein was assigned to the PANTHER (sub)family to which it had the highest hit, if significant. If a protein was assigned to a subfamily, it was also assigned to the full family to which that subfamily belongs. For each PANTHER (sub)family, a phylogenetic profile was constructed and compared to the phylogenetic profiles of the kinetochore proteins. For each kinetochore protein, the best 30 matches of PANTHER (sub)families were selected. The PANTHER protein families often occurring in these top lists can be found in Appendix Table S3.

## Comparing diversity of kinetochore and APC/C proteins

For the kinetochore and APC/C proteins in this dataset, we calculated their occurrence frequencies and entropies across 90 eukaryotic species. The entropy reflects a protein's diversity of presences and absences across species: a protein that is present in half of the species has the highest entropy. We also calculated and compared all pairwise Pearson correlation coefficients of the phylogenetic profiles for both of these protein datasets. To assess how complete the kinetochores and APC/C complexes of the species in our dataset are, we calculated the percentage of present kinetochore proteins in species having Ndc80 and CenpA (because those species are expected to have a kinetochore), and we calculated the percentage of present APC/C proteins in species having the main APC/C enzyme Apc10. Loss frequencies were inferred from Dollo parsimony for all kinetochore and APC/C proteins inferred to have been present in LECA. Transitions (also a measure for the evolutionary dynamics of proteins) were measured for each protein by counting all changes in state (so from present to absent, or from absent to present) along a phylogenetic profile. Since the ordering of the species in the phylogenetic profile is an indication of their relatedness, these transitions are expected to reflect the evolutionary flexibility of proteins as well. dN/dS and percent identity scores for human and mouse sequences were derived from Ensembl [78] (downloaded via Enseml BioMart on November 24, 2016). If multiple one-to-one orthologs for a single orthologous group/family exist, the average dN/dS or percent identity was taken. The results of these kinetochore-APC/C comparisons can be found in Appendix Table S1.

**Expanded View** for this article is available online.

## Acknowledgements

We thank the members of the Kops and Snel labs for critical reading and helpful discussion on the manuscript. We thank John van Dam for his contribution to compiling the eukaryotic proteome database. This work was supported by the UMC Utrecht and the Netherlands Organisation for Scientific Research (NWO-Vici 865.12.004 to GK).

## Author contributions

BS and GJPLK designed the research. JJEH and ET performed the research. LMW contributed the eukaryotic genome database. JJEH, BS, and GJPLK analyzed the data and wrote the manuscript.

## Conflict of interest

The authors declare that they have no conflict of interest.

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
