## [Review Process File · EMBO Reports]

Manuscript EMBO-2017-44102

Evolutionary dynamics of the kinetochore network in eukaryotes as revealed by comparative genomics

Jolien J. E. van Hooff, Eelco Tromer, Leny M. van Wijk, Berend Snel, and Geert J. P. L. Kops

Corresponding author: Geert J. P. L. Kops, Hubrecht Institute

Review timeline:	Submission date:	18 February 2017
	Editorial Decision:	27 March 2017
	Revision received:	22 April 2017
	Editorial Decision:	09 May 2017
	Revision received:	12 May 2017
	Accepted:	17 May 2017

Editor: Achim Breiling

Transaction Report:

1st Editorial Decision

27 March 2017

Thank you for the submission of your research manuscript to EMBO reports. We have now received reports from the three referees that were asked to evaluate your study, which can be found at the end of this email.

As you will see, all three referees highlight the potential interest of the findings. However, all three referees have raised a number of concerns and suggestions to improve the manuscript, or to strengthen the data and the conclusions drawn, which need to be addressed during a revision. As the reports are below, I will not detail them here. In particular, the points by referee #3 need attention, as well as the suggestion of referee #2 to include Snap29 into the analysis.

Given the constructive referee comments, we would like to invite you to revise your manuscript with the understanding that all referee concerns must be addressed in the revised manuscript and in a point-by-point response. Acceptance of your manuscript will depend on a positive outcome of a second round of review. It is EMBO reports policy to allow a single round of revision only and acceptance or rejection of the manuscript will therefore depend on the completeness of your responses included in the next, final version of the manuscript.

REFeree REPORTS

Referee #1:

This interesting and well-written paper by van Hooff, Snel, Kops and colleagues examines the likely presence/absence of kinetochore proteins in a diverse set of 90 eukaryotic organisms. I cannot comment on the quality of the underlying bioinformatics analysis, but the data are certainly of

relevance for researchers interested in the structure and function of kinetochores. Patterns of co-evolution identified by the authors point to functional units, and generally confirm, sometimes revise our current understanding of kinetochore composition, or they point to interesting new candidates for a function at kinetochores. As such, this is a very valuable source of information for researchers in this field.

Minor comments:

- Maybe the authors could indicate in Figure 1A, which proteins they consider having been present in LECA.
 - I assume reference 1 is chosen, because it discusses diverse eukaryotes, but maybe the authors could also point readers to one more review that discusses microtubule/kinetochore connections in more detail.
 - Would it be possible to include RINT-1 and NAG into Figure 2 to demonstrate co-occurrence with ZW10?
 - page 13: A brief explanation of the PANTHER database would be useful.
 - Would it have been possible and informative to include the *T. brucei* KKT proteins in the analysis?
- Language/typos:
- page 6 "Species that contain relatively few...": I think it would be clearer to say "relatively few kinetochore proteins"
 - page 9 "... the CCAN comprises multiple protein complexes..." should probably only refer to Figure 1B (not C).
 - page 12 "Although the TRIP13 phylogenetic profile...": "and" seems too much in the sentence.
 - page 35, legend to Figure 5: "Notable difference exist between...and Cdc20 and Mad1": Is it supposed to be "in Cdc20 and Mad1"? Also, this sentence is currently not clear. If this topic merits a discussion, it should be in the main text; if not, maybe the sentence can be deleted.
 - Supplementary Figure 9: The pink branches are hard to see at this size.

Referee #2:

Our knowledge of kinetochore composition and regulation is mainly based on animals and fungal model organisms, while the diversity of kinetochore complexes across eukaryotes has been addressed in only a few recent studies and reviews. In addition, insights into kinetochore composition of non-model organisms using computational approaches is difficult due the fact that homology-based predictions of kinetochore components can be challenging due to fast rates of protein sequence evolution of several components.

In this study, the authors apply a comprehensive computationally survey predicting 70 kinetochore components as well as proteins of the anaphase promoting complex across 90 different eukaryotes. They apply sensitive HMM-based homology prediction methods taking into account the fast rates of protein sequence evolution. They found that particularly the composition of the kinetochore is dynamic with several components being lost multiple times independently in different eukaryotic lineages (as noted before 10.1038/ncb2493 and 10.3852/15-182). Taking it a step further, they traced back the ancestral composition applying parsimony-based algorithms. They concluded that the ancestral kinetochore likely had a complex composition consisting of around 70% of all identified kinetochore components. Individual kinetochore complexes such as the fungal Dam1 outer complex or the COMA complex likely evolved more recently. While the absence of kinetochore components has been reported in other studies before, their systematic approach allowed the authors to formulate a novel conclusion on a potentially complex composition of an ancestral kinetochore.

Correlating the presence/absence patterns (phylogenetic profile) among each kinetochore component pair showed that some complexes evolve as entities being either completely absent or present in a particular organism. In addition, this approach also allows them to infer the presence of yet unidentified components (recruiters of the RZS complex in organisms without Zwint-1), to deduce information into checkpoint regulation (absence of p31 homolog in select organisms infers absence of a mitotic role of TRIP31 signalling in these organisms), and to predict novel mechanisms for protein interaction while corroborating known interactions (presence of a diversified MIM in Mad1 in land plants and enhanced correlation between the presence of MIM with Mad2).

They further extended their studies to include components of other cellular complexes. Correlating the phylogenetic profiles of spindle assembly checkpoint components with profiles of other components showed novel link to the nuclear pore complex. It will be interesting to explore this link in future experimental analyses as it was done in a previous study constructing phylogenetic profiles for RNAi pathway components and demonstrating a functional link between RNAi and splicing machineries (10.1038/nature11779). Their study could also be extended including Snap29, an unconventional SNARE protein that has recently been demonstrated to localize to the outer kinetochores in flies and humans (10.15252/embj.201693991). Given these data, it would be interesting to include a phylogenetic profile of Snap29 homologs to test whether this interaction is conserved across additional organisms.

Overall, this study is interesting and well executed. This study not only demonstrates the utility of comparative genomics in providing insights into kinetochore plasticity but also provides a very useful resource for the community (particularly because the authors provide all protein sequences in a supplemental file) and paves the way for subsequent experimental follow-up studies in select eukaryotes.

Referee #3:

The kinetochore is required to ensure chromosome segregation throughout eukaryotes. However, the composition of this large structure differs between species and lineages. This paper represents a careful and comprehensive analysis of kinetochore protein conservation across a wide range of eukaryotes. The paper conducts on an extensive amount of homology searching to generate pairwise comparisons for multiple specific proteins, and then uses this data to discuss some significant themes that emerge. For example, the authors discuss the behavior of gains and losses of kinetochore proteins that must have occurred during evolution to result in the observed conservation. They also look at the co-evolution of specific kinetochore protein pairs to reach conclusions about the nature and behavior of kinetochore protein complexes and functions.

This is an interesting paper, and it represents a valuable resource for the kinetochore community. It is not the type of paper that I would typically imagine for EMBO Reports as it is a robust computational analysis of existing data, but does not test any of the predications that the authors make using subsequent experimentation. However, there are a number of important concepts and new ideas that come from this analysis that have the potential to be broadly influential. As such, I support publication of this paper after they address the points below.

1. For their analysis, the authors largely assume that the homology searches will be definitive for identifying the presence of a protein in a given species. However, I think that there is a substantial potential for false negatives indicating the absence of a protein when this is not the case. Kinetochore proteins are not typically constrained by the presence of specific domains (such as a kinase domain) allowing them to evolve more flexibly. Instead, they are characterized by low complexity motifs, such as coiled-coil regions. Because of this, many homologues are at or below the threshold for calling conservation in this type of analysis. For example, the comparison of human and tetrahymena composition in figure 1 seems unlikely to be true as presented. I find it very unlikely that Ndc80 would be conserved, but other members of this complex would not be. Instead, it seems likely that the other subunits have evolved to an extent that puts them below the threshold for detection. If the authors (or another researcher) were to isolate tetrahymena Ndc80, I would anticipate that they would identify 3 proteins corresponding to these missing subunits, which would be detectable based on their features and perhaps a structural comparison.

2. Since the Pearson correlation coefficient is a critical parameter in evaluating the phylogeny profiles, it would be helpful to include a clearer explanation on how this measure and the cut-off value were determined. Specifically, the authors could elaborate on Supplementary Figure 5, and the statistical basis for choosing 6-fold enrichment as the cut-off. Moreover, it would be helpful to clarify the identity of the interacting and non-interacting protein pairs, and how they were selected. If these pairs were within the 70 kinetochore proteins studied, then the parameter and cut-off would be optimized based on only the dataset. As a result, I would be concerned about the predictive power for proteins not included in the study, especially when Pearson correlation is a phylogeny-insensitive measure.

3. One Achilles heel of this type of analysis is that directed experimentation to identify kinetochore components has only been conducted in at most 6-10 organisms. It took 15-20 years or more to arrive at what is likely a near comprehensive list of kinetochore components in the standard model organisms, and it is not feasible to conduct similar approaches in this large range of eukaryotes. Thus, if there are new kinetochore components that arise in evolution (such as the Dam1 complex in fungi), these would not be clear from this type of analysis. There is no way around this, but it would be good for the authors to highlight this point.

4. I found the analysis of duplications to be really interesting. However, as this figure is currently presented, I think that a lot of the key information gets lost. I would be much more interested in seeing the data for cases where there are 2-4 copies of a protein, rather than the cases where there are 60. I would recommend setting the scale to combine all cases with more than 10 copies, and provide a better dynamic range for the colors on the smaller duplication numbers.

5. The authors have constructed a nice set of supplemental files. These are cited in the Methods section, but it may also be helpful to refer to them in selected places in the results where appropriate to highlight these better.

6. I would suggest putting the species names in Figure 1 (as was done in the supplement). It is very helpful to see the details for where proteins are "missing" beyond the broad evolutionary tree.

1st Revision - authors' response

22 April 2017

We hereby submit a revision of our manuscript on comparative genomics of eukaryotic kinetochores (EMBOR-2017-44102). We thank the three reviewers for the generally positive and constructive criticisms, which we hope to have addressed satisfactorily.

In this revised manuscript, we updated the orthologous groups of some of the kinetochore proteins according to our latest insights. These renewed orthologous groups improved the co-evolutionary signal for example for the Mis12 complex, of which the subunits now form a single cluster (Figure 1). In addition, we adopted many of the suggestions put forward by the reviewers. For example, we included the NRZ proteins NAG and RINT1 in our evolutionary analysis of the RZZ complex (Figure 3), which indeed aids in explaining the presences and absences of ZW10 across species. These and all other issues raised are addressed in detail in the point-by-point response attached to this letter.

Point-by-point response (in *italic*) to the referees' comments

Referee: 1

Minor comments:

- Maybe the authors could indicate in Figure 1A, which proteins they consider having been present in LECA.

In the revised version of Figure 1, we indicated the inferred presence of a protein in LECA with a dot.

- I assume reference 1 is chosen, because it discusses diverse eukaryotes, but maybe the authors could also point readers to one more review that discusses microtubule/kinetochore connections in more detail.

In the revised manuscript, we added two additional references that discuss the function and composition of the kinetochore of human and yeast in detail (references 2 and 3).

- Would it be possible to include RINT-1 and NAG into Figure 2 to demonstrate co-occurrence with ZW10?

We thank the reviewer for this suggestion. We determined the orthologous groups of NAG and RINT1 and included the resulting phylogenetic profiles for these proteins in Figure 3 (was Figure 2). Their phylogenetic profiles indeed confirmed our hypothesis that ZW10 mainly co-evolves with NAG and RINT1.

- page 13: A brief explanation of the PANTHER database would be useful.

In the revised manuscript, we describe PANTHER and explained how we used it.

- Would it have been possible and informative to include the *T. brucei* KKT proteins in the analysis?

We originally considered inclusion of KKT proteins, but decided against it. We do not think including them would be informative, because these proteins are only found in kinetoplastids, of which we have only two species in our database. For such rare proteins, our subsequent analysis is unlikely to yield interesting insights. For this reason, we require proteins to be present in at least 3 species of our species set in order to be included in our dataset (see Materials and Methods).

Language/typos:

- page 6 "Species that contain relatively few...": I think it would be clearer to say "relatively few kinetochore proteins"

- page 9 "... the CCAN comprises multiple protein complexes..." should probably only refer to Figure 1B (not C).

- page 12 "Although the TRIP13 phylogenetic profile...": "and" seems too much in the sentence.

- page 35, legend to Figure 5: "Notable difference exist between...and Cdc20 and Mad1": Is it supposed to be "in Cdc20 and Mad1"? Also, this sentence is currently not clear. If this topic merits a discussion, it should be in the main text; if not, maybe the sentence can be deleted.

- Supplementary Figure 9: The pink branches are hard to see at this size.

We apologize for the errors and appreciate the suggestions for improvements. We corrected these errors implemented all suggested changes in the revised manuscript.

Referee: 2

Their study could also be extended including Snap29, an unconventional SNARE protein that has recently been demonstrated to localize to the outer kinetochores in flies and humans (10.15252/embj.201693991). Given these data, it would be interesting to include a phylogenetic profile of Snap29 homologs to test whether this interaction is conserved across additional organisms.

*Upon publication of the study in which Snap29 is identified as a kinetochore protein in *Drosophila*, we had indeed discussed inclusion of its profile in our dataset. However, we decided against it for two reasons: 1) our inclusion criteria required proteins to have a proven kinetochore localization in human cells or budding yeast (the best studied species in this regard). Snap29 seems to play a role during mitosis in human cells, but was not shown to localize to human kinetochores. We thus consider for now Snap29 to be a species-specific kinetochore protein. Ideally, we would want to include all those as well, but this for now is not feasible. 2) Snap29 is a member of the very large family of SNARE proteins, so the resulting phylogenetic profile might not be very informative with respect to its potential kinetochore function.*

Referee 3:

1. For their analysis, the authors largely assume that the homology searches will be definitive for identifying the presence of a protein in a given species. However, I think that there is a substantial potential for false negatives indicating the absence of a protein when this is not the case. Kinetochore proteins are not typically constrained by the presence of specific domains (such as a kinase domain) allowing them to evolve more flexibly. Instead, they are characterized by low complexity motifs, such as coiled-coil regions. Because of this, many homologues are at or below

the threshold for calling conservation in this type of analysis. For example, the comparison of human and tetrahymena composition in figure 1 seems unlikely to be true as presented. I find it very unlikely that Ndc80 would be conserved, but other members of this complex would not be. Instead, it seems likely that the other subunits have evolved to an extent that puts them below the threshold for detection. If the authors (or another researcher) were to isolate tetrahymena Ndc80, I would anticipate that they would identify 3 proteins corresponding to these missing subunits, which would be detectable based on their features and perhaps a structural comparison.

We fully recognize the potential limitations of identifying orthologs by homology searches, particularly because we indeed also noted that various kinetochore proteins evolve rapidly (for example in comparison to APC/C proteins, see Appendix Table S1). Although we performed in-depth and iterative homology searches, our current patterns of presences and absences of kinetochore proteins might indeed contain false negatives. In our initial manuscript, we attempted to address this issue in the Discussion section, but we agree with the reviewer that it is more appropriate that we mention these uncertainties in the Results section. We have therefore amended the first paragraph of this section. Furthermore, we now clarify that our model of the Tetrahymena thermophila kinetochore in Figure 2B (was Figure 1C) is a question mark: our predictions based on homology emphasize the need for more cell-biological knowledge in non-model organisms. The latter we highlight in our revised Discussion.

2. Since the Pearson correlation coefficient is a critical parameter in evaluating the phylogeny profiles, it would be helpful to include a clearer explanation on how this measure and the cut-off value were determined. Specifically, the authors could elaborate on Supplementary Figure 5, and the statistical basis for choosing 6-fold enrichment as the cut-off. Moreover, it would be helpful to clarify the identity of the interacting and non-interacting protein pairs, and how they were selected. If these pairs were within the 70 kinetochore proteins studied, then the parameter and cut-off would be optimized based on only the dataset. As a result, I would be concerned about the predictive power for proteins not included in the study, especially when Pearson correlation is a phylogeny-insensitive measure.

We mainly use the Pearson correlation coefficient as a relative measure (resulting for example in the clustering of proteins in Figure 1, and in the t-SNE map in Appendix Figure S2), making the exact cut-off less important. Nevertheless, we agree that our explanation of why we used the Pearson correlation coefficient instead of other metrics and of our choice of setting the cut-off was insufficient. We hope to have clarified these decisions in the figure legend of Appendix Figure S1 (was Supplementary Figure 5). Indeed, we specifically measured the performances of the different metrics with pairs of kinetochore proteins, and Pearson correlation coefficient was the best one. We do not directly see why kinetochore proteins could be very different from other proteins when measuring co-occurrence, and we reasoned that, since we focus on kinetochore proteins, for this study the best metric was the Pearson correlation coefficient.

3. One Achilles heel of this type of analysis is that directed experimentation to identify kinetochore components has only been conducted in at most 6-10 organisms. It took 15-20 years or more to arrive at what is likely a near comprehensive list of kinetochore components in the standard model organisms, and it is not feasible to conduct similar approaches in this large range of eukaryotes. Thus, if there are new kinetochore components that arise in evolution (such as the Dam1 complex in fungi), these would not be clear from this type of analysis. There is no way around this, but it would be good for the authors to highlight this point.

This is indeed an essential limitation of these types of analyses. We elaborate on this limitation in the Discussion section. We expect that characterization of other species' kinetochores might greatly enhance our understanding of kinetochore diversity and evolution.

4. I found the analysis of duplications to be really interesting. However, as this figure is currently presented, I think that a lot of the key information gets lost. I would be much more interested in seeing the data for cases where there are 2-4 copies of a protein, rather than the cases where there are 60. I would recommend setting the scale to combine all cases with more than 10 copies, and provide a better dynamic range for the colors on the smaller duplication numbers.

We thank the reviewer for this helpful suggestion, which we have adopted in Figure EV 3.

5. The authors have constructed a nice set of supplemental files. These are cited in the Methods section, but it may also be helpful to refer to them in selected places in the results where appropriate to highlight these better.

The format of EMBO Reports allows for including Source Data with figures, which provided us an excellent means to directly incorporate the orthologous sequences as data underlying Figure 1, and thereby to include it in the main manuscript. In addition, in the revised manuscript we reported the availability of these sequences in the first paragraph of the Results section.

6. I would suggest putting the species names in Figure 1 (as was done in the supplement). It is very helpful to see the details for where proteins are "missing" beyond the broad evolutionary tree.

We have now included the complete species names in Figure 1.

2nd Editorial Decision

09 May 2017

Thank you for the submission of your revised manuscript to our editorial offices. We have now received the reports from the three referees that were asked to re-evaluate your study that you will find enclosed below. As you will see, all three referees now support the publication of your manuscript in EMBO reports. Referee #3 has one further comment that we ask you to address in a final revised version. Further, I have these editorial requests:

Please format the references according to EMBO reports style. See:
<http://embor.embopress.org/authorguide#referencesformat>

Could you please check for all figures if they are conforming to our guidelines?
 See: http://embopress.org/sites/default/files/EMBOPress_Figure_Guidelines_061115.pdf

Figures should not have landscape format. In Figure EV2 the writing of the gene names is too small. Could you provide a version with bigger fonts? Then please upload all figures as editable TIFF or EPS-formatted single figure files in high resolution (for main figures and EV figures).

I look forward to seeing a revised version of your manuscript when it is ready. Please let me know if you have questions or comments regarding the revision.

REFeree REPORTS

Referee #1:

The authors have adequately answered all my previous comments. The changes to the figures are useful and make it easier to follow the arguments in the text. The text has also been improved.

In my opinion, the study can be published in the current form. I expect that many researchers in the community will find the pieces of information that are highlighted very useful, and will also mine this comprehensive data set for additional information.

Referee #2:

The authors properly answered to the requests and the paper is now suitable for publication in EMBO reports.

Referee #3:

This revised manuscript is further improved. Based on the changes, I enthusiastically recommend publication in EMBO Reports. I have only one very minor comment that the authors could revisit:

In the figure showing the copy number for various kinetochore components across species, the new color-coding is a significant improvement. However, as the results for this are now clearer, I was

able to spot-check some of the information. From my quick look, there appear to be some errors. For example, several proteins are listed as being duplicated (or more) in humans for which I am unable to find any evidence for this based on the literature or BLAST searches (Astrin, Ska1, Ska2, Spc25, Nnf1). Perhaps I am incorrect (in which case these are quite exciting), or maybe there has been an error in assembling this table.

2nd Revision - authors' response

12 May 2017

We hereby submit a second revision of our manuscript on comparative genomics of eukaryotic kinetochores (EMBOR-2017-44102).

In addition to the editorial requests, in this revised manuscript we addressed the comment made by referee #3, which we further elaborate on below:

“... In the figure showing the copy number for various kinetochore components across species....From my quick look, there appear to be some errors. For example, several proteins are listed as being duplicated (or more) in humans for which I am unable to find any evidence for this based on the literature or BLAST searches (Astrin, Ska1, Ska2, Spc25, Nnf1). Perhaps I am incorrect (in which case these are quite exciting), or maybe there has been an error in assembling this table.”

We case-by-case verified all human genes (including the ones mentioned by the referee) in Ensembl (May 10, 2017), the genome annotation system from which we derived the human proteome in our local database. Ensembl removed some of the entries that our local database still contained (Astrin, Spc25), and it included some entries that actually mapped to the same genomic position, on the same strand (Astrin, Ska1, Nnf1), due to which these should be considered single genes. These findings confirmed that proteome databases, including the ones we incorporated in our local database, may contain errors, leading both to underestimations of gene copy numbers as well as to overestimations. We corrected the copy numbers of these human genes in Figure EV3 and removed the redundant protein sequences in the supplied fasta files. In addition, we point out the uncertainties in the caption of Figure EV3.

We look forward to your response and of course we are willing to answer any additional questions that may arise.

3rd Editorial Decision

17 May 2017

I am very pleased to accept your manuscript for publication in the next available issue of EMBO reports. Thank you for your contribution to our journal.

Corresponding Author Name: Geert Kops

Journal Submitted to: EMBO reports

Manuscript Number: EMBO-2017-4102